# Ultrastructural and Molecular Investigation on Peripheral Leukocytes in Alzheimer’s Disease Patients

**DOI:** 10.3390/ijms24097909

**Published:** 2023-04-26

**Authors:** Roberta Giannelli, Paola Canale, Renata Del Carratore, Alessandra Falleni, Margherita Bernardeschi, Francesca Forini, Elisa Biagi, Olivia Curzio, Paolo Bongioanni

**Affiliations:** 1Institute of Clinical Physiology, National Research Council, 56124 Pisa, Italy; 2Department of Experimental and Clinical Medicine, University of Pisa, 56126 Pisa, Italy; 3Italian Institute of Technology, Center for Materials Interfaces, Smart Bio-Interfaces, 56025 Pontedera, Italy; 4BMS Multispecialistic Biobank-Biobank Unit, AOUP—Pisa University Hospital, 56126 Pisa, Italy; 5Severe Acquired Brain Injuries Dpt Section, Azienda Ospedaliero Universitaria Pisana, 56100 Pisa, Italy; 6NeuroCare Onlus, 56100 Pisa, Italy

**Keywords:** Alzheimer’s disease, neutrophils, ultrastructure, endoplasmic reticulum, neurodegeneration, blood biomarkers

## Abstract

Thriving literature underlines white blood cell involvement in the inflammatory processes of Alzheimer’s Disease (AD). Among leukocytes, lymphocytes have been considered sentinels of neuroinflammation for years, but recent findings highlighted the pivotal role of neutrophils. Since neutrophils that infiltrate the brain through the brain vascular vessels may affect the immune function of microglia in the brain, a close investigation of the interaction between these cells is important in understanding neuroinflammatory phenomena and the immunological aftermaths that follow. This study aimed to observe how peripheral leukocyte features change at different stages of AD to identify potential molecular markers when the first features of pathological neurodegeneration arise. For this purpose, the examined patients were divided into Mild Cognitive Impairment (MCI) and severely impaired patients (DAT) based on their Cognitive Dementia Rating (CDR). The evaluation of the neutrophil-to-lymphocytes ratio and the morphology and function of leukocytes showed a close relationship between the ultrastructural and the molecular features in AD progression and suggested putative markers for the early stages of the disease.

## 1. Introduction

Burgeoning evidence implicates the involvement of leukocytes in neuroinflammation in AD pathogenesis and progression, predominantly involving the innate immune system [1]. Neuroinflammation may be initiated in response to a variety of stimuli such as viral or bacterial infection, traumatic injury, toxic metabolites, autoimmunity, stress, aging, and lifestyle [2,3]. In a healthy brain, neuroinflammation, which is mainly controlled by resident mediators of the immune response, i.e., microglia and astrocytes, contributes to maintaining local homeostasis, the secretion of neurotrophic factors, the removal of cell debris, and the release of pro-inflammatory cytokines [4]. When the immune response is no longer under control, it becomes detrimental for neurons and leads to chronic inflammation, which is causative of neurodegenerative disorders [5,6]. For years, lymphocytes have been considered as sentinels of neuroinflammation [7], but recent findings highlighted the pivotal role of neutrophils [8], since they are the first-line defenses enrolled in the innate immune response and in the resolution of acute inflammation [1,9]. Under physiological conditions, neutrophils contribute to damage resolution and tissue repair through membrane extroversions—pseudopods [10]—by phagocytizing necrotic cells and stopping them from attracting more immune cells. They release mediators to promote growth and angiogenesis [11] and constitutively undergo apoptosis after approximately 24 h [12], whereas during inflammation apoptosis can be delayed by cytokines, such as Tumor Necrosis Factor-alpha (TNF-α) and Granulocyte-Macrophage Colony-Stimulating Factor (GM-CSF) [13].

Several studies demonstrated that AD patients present a higher number of neutrophils in the peripheral blood than controls [14]. Indeed, during disorders or infections of the central nervous system (CNS), microglia are activated and thus undergo morphological changes, such as enlargement of the cell body and shortening of processes. Such activated microglia generate chemoattractants that induce neutrophil infiltration into the brain. The following interaction between these two cell types potentially induces neutrophil molecular modifications. Thereafter, neutrophils are engaged in reverse transendothelial migration, returning to the bloodstream [15]. Although these events make leukocyte peripheral biomarkers carry important information from the CNS, few studies have been reported about such a role to date.

Our investigation aimed to evaluate whether alterations in the amount, morphology, and molecular patterns of peripheral leukocytes, with a focus on neutrophils, may be related to the early symptoms of Alzheimer’s Disease.

## 2. Results

### 2.1. NLR and Related Molecular Markers

The neutrophil-to-lymphocyte ratio (NLR) was calculated in each group of subjects (CTR, MCI, DAT) as described in Methods. The neutrophil number significantly increased, while that of lymphocytes decreased in MCI and DAT with respect to CTR (Figure 1). However, a reverse trend appeared in DAT as the disease worsened.

Because a survival-promoting effect on neutrophils is known to be activated by TNF-α through an NF-kB-dependent pathway and its inhibitor NFKBIA [14], their levels were tested by qRT-PCR (Figure 2A-C). A significant increase of TNF-α and NF-kB was observed in MCI patients followed by a decrease as the disease worsened (Figure 2A,B), whereas NFKBIA increased in a more advanced phase of the disease (Figure 2C).

### 2.2. Leukocyte Analyses by Transmission Electron Microscopy

The leukocytes from each group (CTR, MCI and DAT) were analyzed at the ultrastructural level to evaluate the presence of pseudopods and autophagic vacuoles and possible alterations in the endoplasmic reticulum and mitochondria.

The leukocyte population was composed of neutrophils, lymphocytes, monocytes, eosinophils, and basophils (Figure 3A–C). The evidence of thin pseudopods on the neutrophil surface of MCI and DAT patients (Figure 3B,C) suggested their active state. The number of pseudopods on the neutrophil surface was significantly increased in MCI and DAT patients with respect to the controls (Figure 3D).

Many diseases are related to mitochondrial and RER distress. Comparing micrographs from MCI patients, DAT patients, and healthy CTR, striking alterations in RER and mitochondria were detected associated with AD outcome.

In healthy controls, the RER consisted of long normally structured cisternae with a medium-electron-density content (Figure 4A), while in MCI and DAT patients, RER cisternae were shorter and enlarged, with a less dense content (Figure 4B–D). As shown in the histogram (Figure 4E), the percentage of leukocytes with altered RER was significantly higher in MCI and DAT patients with respect to the controls, and a statistical difference was also observed between MCI and DAT patients.

In healthy controls, the mitochondria showed an electron-dense matrix and well-arranged cristae (Figure 5A and Figure 6A); in MCI and DAT patients, the mitochondria appeared as poorly electron-dense hypertrophic organelles showing matrix dilution, with occasionally disarranged and/or broken cristae (Figure 5B,C). The percentage of cells with altered mitochondria was significantly increased in MCI and DAT patients with respect to the controls (Figure 5D).

Autophagic vacuoles (AVs) were selected as a matter of interest since, although their formation is a regular process in the cell, an increase in their number may be a sign of a dysfunctional state. As reported in the Methods section, AVs were identified as double-membrane-bound structures containing altered cytoplasmic organelles or cytosolic material (Figure 6A–C). Although no statistically significant differences were observed when comparing the three groups of subjects, a trend of increase in the number of AVs was found in MCI patients (Figure 6D).

### 2.3. Molecular Markers Associated with Morphological Changes

The morphological alterations observed in the leukocyte samples by TEM prompted us to test the expression of well-known markers of RER stress, mitochondrial function, and autophagy by the analysis of the level of mRNAs through qRT-PCR [16].

As main mediators of RER stress, we evaluated PKR-like ER kinase (*PERK*), activated transcription factor 6 (*ATF6*), X-box binding protein 1 (*XBP*1), and C/EBP homologous protein (*CHOP*). The results showed that *PERK*, *XBP1*, and *CHOP* expression levels were significantly higher in MCI patients with respect to CTR, while tending to return to CTR levels in DAT (Figure 7A,C,E). Conversely, the expression of ATF6 progressively increased with disease severity (Figure 7B).

Concerning mitochondrial impairments, Bcl-2-Associated X-protein (*BAX*) was chosen as a marker, given its critical role as a gateway to mitochondrial dysfunction and cell death [17]. As shown in Figure 7D, BAX level was significantly higher in DAT versus CTR, while a trend to increase was observed in MCI.

Finally, the level of autophagy-related gene 5 (*ATG5*), which is considered a marker of autophagy, was significantly increased in MCI patients (Figure 7F).

## 3. Discussion

In this work, we followed AD patients during disease evolution, assessing the number and the ultrastructure of leukocytes, with particular attention to the neutrophil subpopulation to understand if leukocytes may be carriers of information relating to AD onset in order to have easily available early markers. Our data confirmed significantly elevated neutrophil and leukocyte counts and a higher NLR in MCI patients compared with healthy controls. Despite NLR being considered a marker of many inflammatory conditions, including rheumatoid arthritis, inflammatory bowel disease, hepatitis, vasculitis, infectious processes and cancer [14,18], the interpretation of its significance remains controversial [19]. Different studies demonstrated that also AD patients’ peripheral blood contains an increased number of neutrophils [14,18], which has been associated with increased amyloid burden, but this association was no longer present after adjustment for sex, apolipoprotein E4, and mainly age, thus limiting the use of NLR as a diagnostic biomarker in more advanced stages of the disease [19]. A limit of the experimentation on neurodegenerative patients is generally their great heterogeneity, i.e., comorbidities, different ages, and the complex physiological-clinical pattern.

Starting with these assumptions, we collected in a database the clinical (CDR) and biochemical data related to each blood withdrawal that patients underwent diachronically.

In such a way, we could make a very careful choice of patients by excluding people with other neurological and inflammatory processes which might interfere with the experimental results.

In our approach, we assayed NLR values in MCI and DAT patients separately, observing a significatively enhanced NLR in MCI and DAT patients as compared to CTR. These results suggest NLR as a putative marker already in the early onset of the illness and, in combination with the fact that NLR did not further increase in DAT, they seem to exclude the influence of aging.

Another critical issue raised by various researchers is that NLR is generally calculated when the patients’ CDR is at least 1, since before it is not possible to distinguish an AD onset from a physiological cognitive decline due to aging [14]. This study aimed to observe how peripheral leukocyte features change mainly at the onset of AD to identify potential molecular markers when the first events of pathological neurodegeneration occur. We collected blood samples from patients with slight cognitive impairment, functionally indistinguishable from those of age-matched control subjects, and stocked them in a biobank even several years before the patient was diagnosed with a neurodegenerative disorder. This allowed us to analyze retrospectively the samples when the disease was established and, thus, focus on the first events occurred as the illness manifested.

The increase of NLR in MCI patients might be related to the inflamed brain environment which induces resident microglia to produce pro-inflammatory cytokines, recruit neutrophils, and cause their activation in the blood (Figure 8), as reported by Park et al. [20] and Hijioka et al. [21].

More precisely, TNF-α produced by microglia is known to exert a survival-promoting effect on neutrophils via the release of IL-9 through an NF-kB-dependent pathway [14]. The elevated level of NF-kB we found in our MCI patients confirmed this event that persisted until the damage was so severe that the therapeutic mechanism stopped working, and the NLR value returned to the physiological levels, as reported by Schramm et al. [22].

Since leukocytes exposed to the inflamed brain environment after contact with microglial cells acquire specific features [15], important information could be obtained by their morphological and functional examination. During neutrophil–microglial cross-talk, microglial processes were observed to stretch toward the point of contact, assuming an amoeboid morphology [15]. Such events reinforce the idea that contact between leukocytes (neutrophils) and microglia accompanies intercellular communication, affecting not only the interacting cells but also the surrounding environment. The presence of pseudopods we observed on the neutrophil surface could be further proof of their activation and, consequently, of their potential as biomarkers as the early symptoms of neurodegeneration appear.

Ultrastructural analyses performed on leukocytes showed a remarkably increased RER stress and mitochondrial alterations occurring in MCI and DAT patients with respect to the controls. In addition, concerning the RER stress, the highest percentage of organelle abnormalities observed in MCI patients is in line with the critical role recently attributed to RER impairments in AD onset [23].

The control of protein synthesis and folding operated by the RER is important in brain development, network connection, and synapse function. Although the RER has been known to play a role in protein synthesis, now it is considered to be also involved in a wide range of biochemical processes, such as lipid metabolism or Ca^2+^ homeostasis [24,25]. Cellular proteins must be folded and post-translationally modified by a favorable ER environment. In stressful situations, RER luminal conditions or chaperone capacity are altered, and the cell activates signaling cascades to restore the appropriated folding environment, triggering the so-called unfolded protein response (UPR) that can lead to autophagy to preserve cell integrity. These processes are mainly regulated by three factors: *PERK*, *ATF6*, and *IRE1* [26], as described in Figure 9.

*PERK* indirectly induces the activating transcription factor 4 (*ATF4*) which translocates to the nucleus promoting *CHO*P expression. *ATF6* translocates to the Golgi compartment, where it is cleaved and activated and serves as a transcription factor for both *CH*OP and *XBP-*1. Similarly, *XBP1*, activated by *IRE1*, translocates to the nucleus promoting *CHOP* transcription [26]. In physiological conditions, *CHOP* is ubiquitously expressed at very low levels, whereas under pathological conditions which cause stressed RER, the expression of *CHO*P is upregulated.

The *CHOP* signaling pathway is considered to play a pivotal function in inducing cell apoptosis by upregulating the expression of death receptor 4 (DR4) and death receptor 5 (DR5), which lead to the extrinsic apoptotic pathway. CHOP can also cause increased *BAK* and *BAX* expression that leads to the release of apoptotic factors (intrinsic pathway) such as cytochrome c (Cyt-C) and apoptosis-inducing factor (AIF) through mitochondria permeabilization and cause cell death [26].

These assumptions agree with our molecular data showing increased *PERK*, *XBP1*, and *CHOP* expression levels in MCI patients and enhanced *ATF6* and *BAX* in DAT patients, suggesting a pronounced RER stress at the onset of dementia, potentially followed by apoptosis at later stages of the disease.

Based on our results and considering that the presumed contact of leukocytes with an inflamed brain environment may alter their morphology, the stressed RER may be a signal of neurodegeneration for AD, as also suggested by Merighi et al. [25]. Indeed, RER stress has been demonstrated to be involved in different neurodegenerative diseases, such as Parkinson’s Disease and Amyotrophic Lateral Sclerosis [25]. Therefore, we propose to distinguish AD onset from other neurological disorders based on an integration of RER stress with more specific clinical traits. In addition, in our work we propose to use RER stress in peripheral leukocytes as a biomarker to identify AD onset as compared with age- and sex-matched healthy controls, after having excluded other neurodegenerative conditions.

Because of their involvement in many pathological conditions, including neurodegenerative processes, autophagic vacuoles (AVs) were evaluated as well. Even though no significant difference was observed in our three groups of subjects, an increasing trend in the number of AVs was found in MCI patients. This tendency was confirmed, at the molecular level, by the increased expression of *ATG5*, a specific marker of the autophagy pathway. Furthermore, the fact that there were no significant differences in the number of autophagic vacuoles among the three groups and that the expression level of *ATG5*, the master regulator of autophagy, was significantly upregulated in MCI patients, could induce to speculate the existence, also in leukocytes, of a secretory autophagy pathway (unconventional non-degradative process) involved in the recruitment of the cargo and the secretion of extracellular vesicles (EVs), as it has recently been reported in other mammalian cells [27,28]. Moreover, at the disease onset in MCI patients, the EVs produced by leukocyte secretory autophagy could contribute to the activation of further pathogenetic mechanisms and therefore lead to AD progression.

Overall, while the mechanisms underlying the pathophysiology of AD are yet to be revealed, these results strongly suggest not only that inflammatory states promote crosstalk and contact between resident and peripheral immune cells, but also that the latter spread proinflammatory signals to peripheral compartments outside the brain [15].

Taken collectively, the morphological and molecular biomarkers observed in peripheral blood cells reported in our work may represent early functional and diagnostic indices for Alzheimer’s disease.

## 4. Materials and Methods

### 4.1. Patients Enrolled

AD patients, including 51 men (80 ± 4.5 y.o.) and 84 women (84 ± 3.8 y.o.) in charge of the local hospital AOUP (Azienda Ospedaliero Universitaria Pisana), were diachronically followed over the past 8 years (from 2014 to 2022). Their clinical data, such as the Clinical Dementia Rating (CDR) score and biochemical values (106 blood analytes), were recorded in a database, as described more in detail by Baldini et al. [29]. Each patient was registered with a code and information about their age, sex, and disease onset. The patients were grouped into mild cognitive impairment (MCI) subjects with CDR = 0.5–1 and severely impaired patients (DAT) with CDR = 2–5. Age- and sex-matched subjects were selected as controls (CTR). Written informed consent was obtained from all the patients, and the regional Human Ethics Committee approved the study (n°14568). Patients with inflammatory comorbidities and other associated neurological disorders (such as Parkinson’s disease and Amyotrophic Lateral Sclerosis), that could interfere with the results, were excluded from our analysis.

### 4.2. Analysis of the Neutrophil-to-Lymphocyte Ratio (NLR)

The neutrophil-to-lymphocyte ratio (NLR) was calculated by dividing the absolute number of neutrophils by the number of lymphocytes. To this aim, the White Blood Cells (WBC) formula contained in the database for each patient was used, considering the group division based on the CDR (MCI and DAT). A total of 388 observations were collected as follows: 90 for CTR, 103 for MCI, and 195 for DAT (Table 1).

### 4.3. Leukocyte Preparation

The blood samples were diluted 1:1 with PBS, mixed carefully, and gently layered to 4 mL with Lympholyte solution (Cedarlane, Burlington, VT, Canada) in a conic 15 mL tube. Lympholyte solution was used as a density gradient separation medium for the isolation of mononucleated cells from the other blood cells.

The samples were centrifuged at 800× *g* for 20 min, and the interphase containing leukocytes was carefully aspirated; 3 volumes of PBS were added to the leukocytes, centrifugation was performed at 120× *g* for 10 min, and the supernatant was removed. The cell pellets were suspended in 500 μL of PBS, and 20% DMSO and 80% human albumin were added. A SyLab IceCube 1810 Cd or SyLab programmed descent freezer was used for vital leukocyte cryopreservation.

### 4.4. Transmission Electron Microscope Analysis

Five MCI patients, five DAT patients, and five age-matched CTR were enrolled for leucocyte examination by transmission electron microscope (TEM). The pellets from each experimental group were washed twice in PBS and fixed in 2.5% glutaraldehyde in 0.1 M cacodylate buffer, pH 7.2, for 2 h at 4 °C, washed in the same buffer, and postfixed in 1% osmium tetroxide in 0.1 M cacodylate buffer for 1 h at room temperature. After dehydration in graduated series of ethanol, the samples were transferred into propylene oxide for 15 min, finally embedded in Epon-Araldite, and polymerized at 60 °C. Ultrathin sections (60–90 nm), obtained with a Reichert–Jung Ultracut E (Rechert-Jung, Wien, Austria) equipped with a diamond knife, were collected on 200-mesh formvar/carbon-coated copper grids, double stained with aqueous uranyl acetate and lead citrate, and examined with a Jeol 100 SX transmission electron microscope (Jeol, Tokyo, Japan) operating at 80 kV. Micrographs were obtained with an AMTXR80b Camera System.

### 4.5. Morphometric Analysis: Assessment of Rough Endoplasmic Reticulum, Mitochondria, Autophagic Vacuoles, and Pseudopods

In our experimental setting, we took advantage of cellular pellets because ultrathin sections contain randomly oriented cells, which is an appropriate condition for quantitative calculations [30,31]. For ultrastructural morphometry, non-serial ultrathin sections were obtained for each experimental group as described above. The ultrathin sections were examined directly at the transmission electron microscope at 4000–6000× magnification to estimate the status of mitochondria (altered/normal), RER cisternae (altered/normal), and the number of autophagic vacuoles (AV) and neutrophil pseudopods per cell. In particular, for mitochondria membrane integrity, matrix dilution, cristolysis, and organelle hypertrophy were evaluated; for the RER evaluation, enlargement, vesiculation, altered content density of the cisternae together with ribosomal derangement or disjunction were analyzed. AVs were identified as double-membrane-bound structures (compartments) containing altered cytoplasmic organelles or cytosolic material. At least 5 grids were examined for each experimental group, and 50 cells were scored for each sample. Score = 1 was assigned if the mitochondria and RER were altered, whereas score = 0 was assigned when the mitochondria and RER were not altered. The number of AVs and pseudopods was counted for each cell.

### 4.6. Quantification of mRNA by qRT-PCR

Molecular markers associated with morphological alterations were detected by qRT-PCR. Total RNA was extracted from the leukocyte samples with the miRNeasy mini kit reagent (Qiagen, Hilden, Germany) according to the manufacturer’s instructions. RNA quality and amount were determined using a Nanodrop TM Lite Spectrophotometer (Thermo Fisher Scientific, Waltham, MA, USA) and by gel electrophoresis. For the analysis, 500 ng of total RNA was retrotranscribed using the iScriptTM cDNA Synthesis Kit (BioRad, Hercules, CA, USA) following the manufacturer’s instructions. For gene expression analysis, 5 ng of cDNA was processed in duplicate in a Rotor-Gene Q real-time machine (Qiagen, Hilden, Germany) using the SsoAdvanced Universal SYBR^®^ Green SuperMix (BioRad, Hercules, CA, USA). The PCR conditions were as follows: 30 s of initial denaturation (95–98°) and then 40 cycles at 95 °C for 5 s, 60 °C for 20 s. To assess product specificity, a melting curve analysis from 65 °C to 95 °C was performed. The gene transcript values were normalized using GAPDH as a reference gene. The relative quantification of the samples was performed by the Rotor-Gene AssayManager^®^ v1.0 (Qiagen, Hilden, Germany). The complete list of the primer sequences is shown in Table 2.

### 4.7. Statistical Analysis

For the morphometric analysis by TEM, parametric variables were presented as mean ± DS. The data followed a Bernoulli binomial distribution, as explained in “Morphometric analysis: assessment of rough endoplasmic reticulum, mitochondria, autophagic vacuoles, and pseudopods”. The comparison among groups was performed by the multifactor analysis of variance, MANOVA. The multiple range test (MRT) was performed to detect differences among the experimental groups. For all data analysis, statistical significance was set at *p* < 0.05.

For qRT-PCR, we verified the distribution of the parameters for normality by the Shapiro–Wilk test before inferential statistical analysis. All parametric variables are presented as mean ± SEM. Differences between the means of the three groups were evaluated by ANOVA followed by the Least Significant Difference (LSD) post hoc test for multiple comparisons (IBM SPSS Statistics, Armonk, NY, USA, version 26). The results were considered statistically significant with *p* ≤ 0.05 values. For non-parametric variables, we used the Kruskal–Wallis test. Thereafter, the Mann–Whitney U-test was used to check differences between groups two by two, adjusting the α-level by Bonferroni inequality. The differences were considered statistically significant at a value of *p* ≤ 0.017.

## 5. Conclusions

Our investigation addressed the question of whether peripheral blood cells carry the hallmarks of a neuroinflammatory state in AD patients. Our results, obtained by TEM and confirmed by related molecular markers evaluation, highlight the potential role of peripheral leukocytes in revealing potential neuroinflammatory events at the onset of AD. As a consequence, these achievements may expand the burden of further speculations on the pathophysiology of AD.

## Figures and Tables

**Figure 1 ijms-24-07909-f001:**
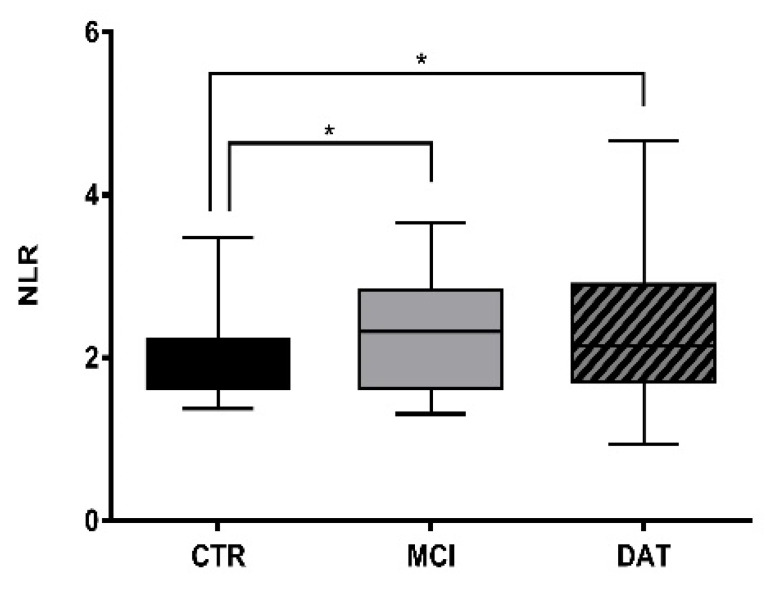
NLR comparison among groups. Boxplot representing NLR calculated in controls and patients, with 51 MCI patients (CDR = 0.5–1), 84 DAT patients (CDR = 2–5), and 45 healthy controls (CTR). Kruskal–Wallis non-parametric test followed by Mann–Whitney U-test for group comparison * *p* < 0.05.

**Figure 2 ijms-24-07909-f002:**
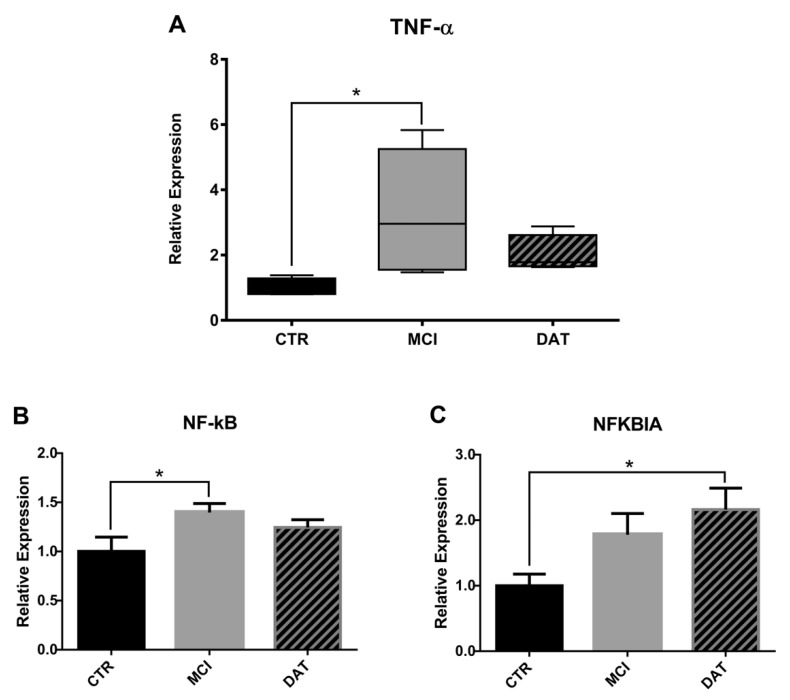
Relative expression analysis of molecular markers relevant for neutrophil survival. TNF-α (**A**), NF-kB (**B**), and NFBIA (**C**) were assayed in leukocytes by qRT-PCR in five MCI and five DAT patients with respect to five healthy controls (CTR). (**A**) Kruskal–Wallis non-parametric test followed by Mann–Whitney U-test for group comparison with * *p* < 0.017 and (**B**,**C**) ANOVA parametric test followed by the Least Significant Difference (LSD) post hoc test for multiple comparisons. * *p* < 0.05.

**Figure 3 ijms-24-07909-f003:**
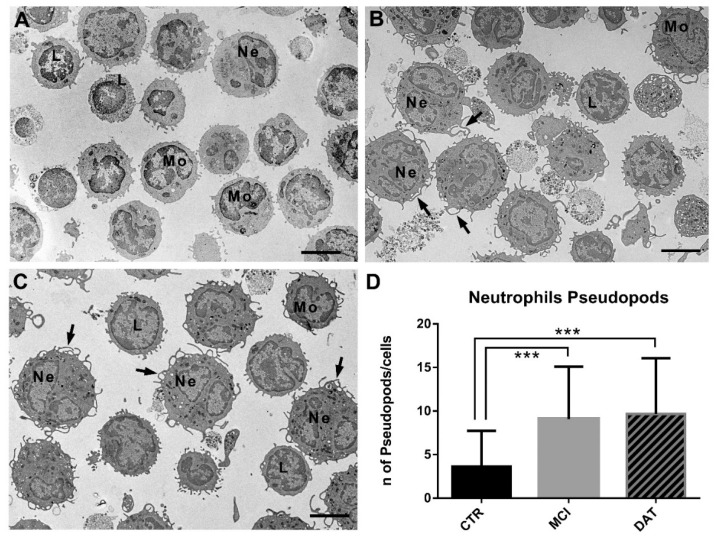
Representative TEM images of leukocytes showing pseudopods on the cell surface. (**A**) Control leukocytes with few cell membrane protrusions. (**B**) MCI and (**C**) DAT leukocytes, in particular neutrophils, showed a high number of thin pseudopods (arrows) on the cell surface. Scale bars, 4 µm. L, lymphocyte; Mo, monocyte; Ne, neutrophils. (**D**) The number of pseudopods on the neutrophil surface was significantly higher in MCI and DAT patients with respect to the controls. MANOVA followed by multiple range test (MRT). *** *p* < 0.001.

**Figure 4 ijms-24-07909-f004:**
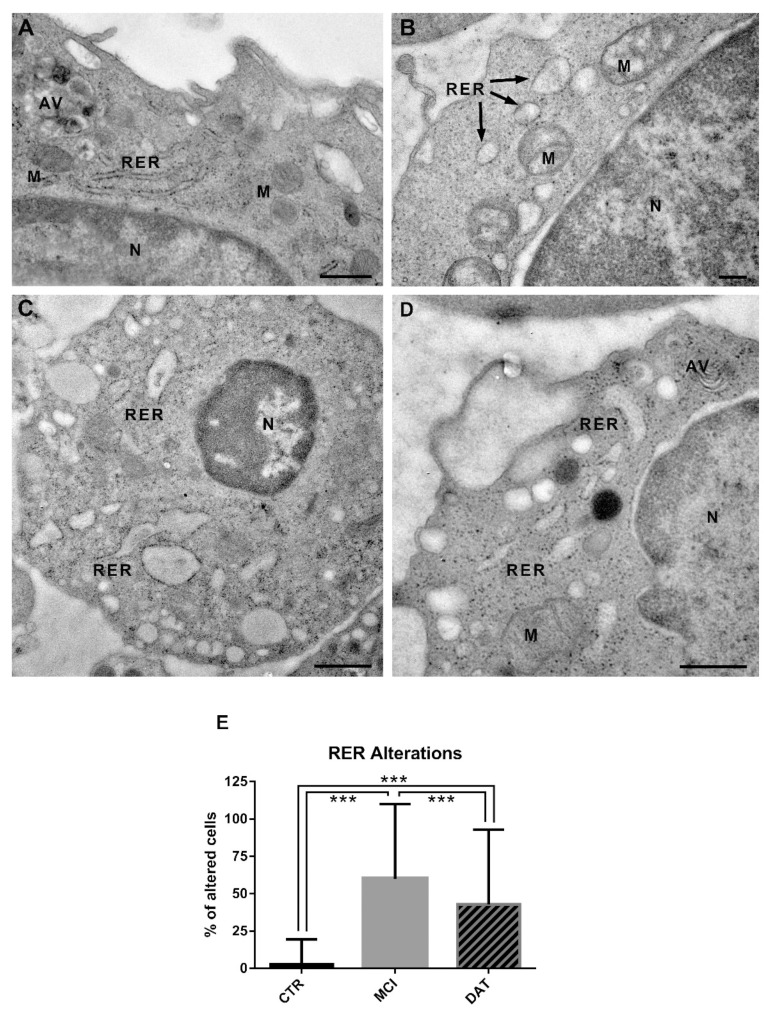
Representative TEM images of the rough endoplasmic reticulum (RER) in leukocytes. (**A**) Control leukocytes showing long and well-structured cisternae of the RER. Scale bar, 600 nm. (**B**,**C**) Leukocytes from MCI patients showing an altered RER consisting of short, enlarged cisternae. Scale bars, (**B**) 200 nm; (**C**) 800 nm. (**D**) leukocytes from DAT patients with short and swollen RER cisternae. Scale bar, 500 nm. AV, autophagic vacuole; M, mitochondrion; N, nucleus. (**E**) The percentage of leukocytes with altered RER cisternae was significantly increased in MCI and DAT patients with respect to the controls. In addition, a statistically significant difference was highlighted between MCI and DAT patients. MANOVA followed by multiple range test (MRT). *** *p* < 0.001.

**Figure 5 ijms-24-07909-f005:**
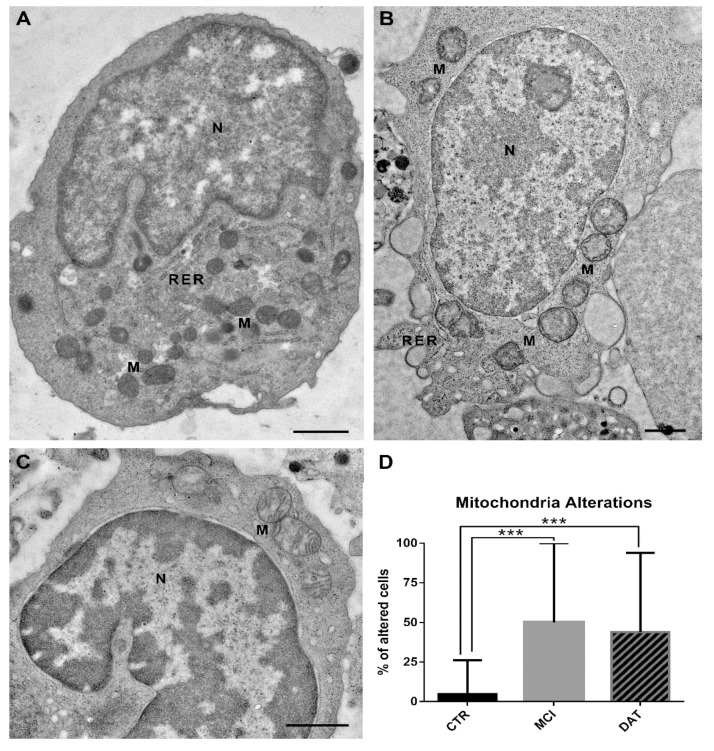
Representative TEM images of mitochondria (M) in leukocytes. (**A**) Control leukocytes showing mitochondria with a dense matrix and well-organized cristae. (**B**) MCI and (**C**) DAT leukocytes depicting swollen mitochondria with a diluted matrix and disarranged/broken cristae. Scale bars, 1 µm. N, nucleus; RER, rough endoplasmic reticulum. (**D**) The percentage of cells with altered mitochondria was significantly increased in MCI and DAT patients with respect to the controls. MANOVA followed by multiple range test (MRT). *** *p* < 0.001.

**Figure 6 ijms-24-07909-f006:**
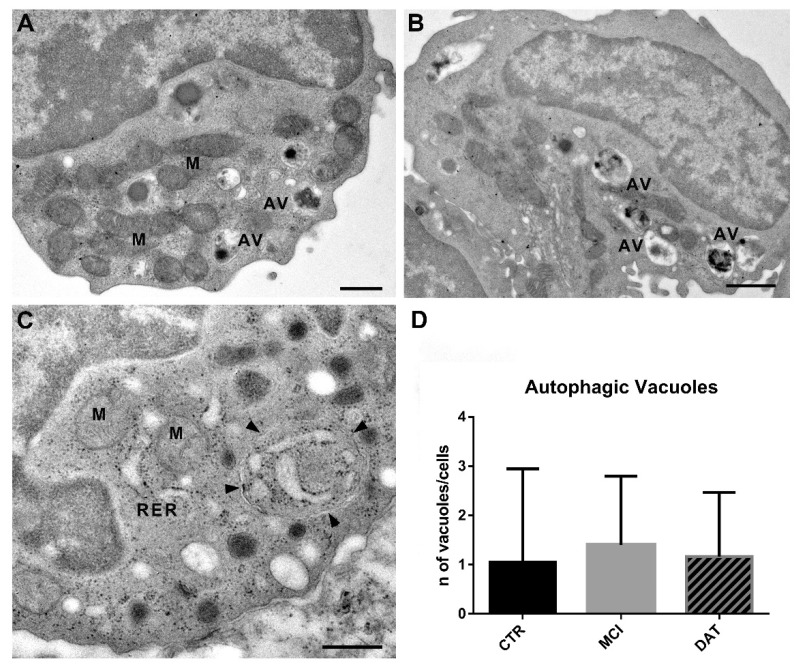
Representative TEM images of autophagic vacuoles (AVs) in leukocytes. (**A**) Control leukocytes showing few AVs and well-structured mitochondria (M). Scale bar, 600 nm. (**B**) Leukocytes from MCI patients with an increased number of AVs. Scale bar, 1 µm. (**C**) Leukocytes from DAT patients. Arrowheads point to an autophagic vacuole delimited by a double membrane containing swollen cisternae of RER. Scale bar, 500 nm. M, mitochondrion; RER, rough endoplasmic reticulum. (**D**) Histogram showing the number of AVs in the three different groups. MANOVA followed by multiple range test (MRT).

**Figure 7 ijms-24-07909-f007:**
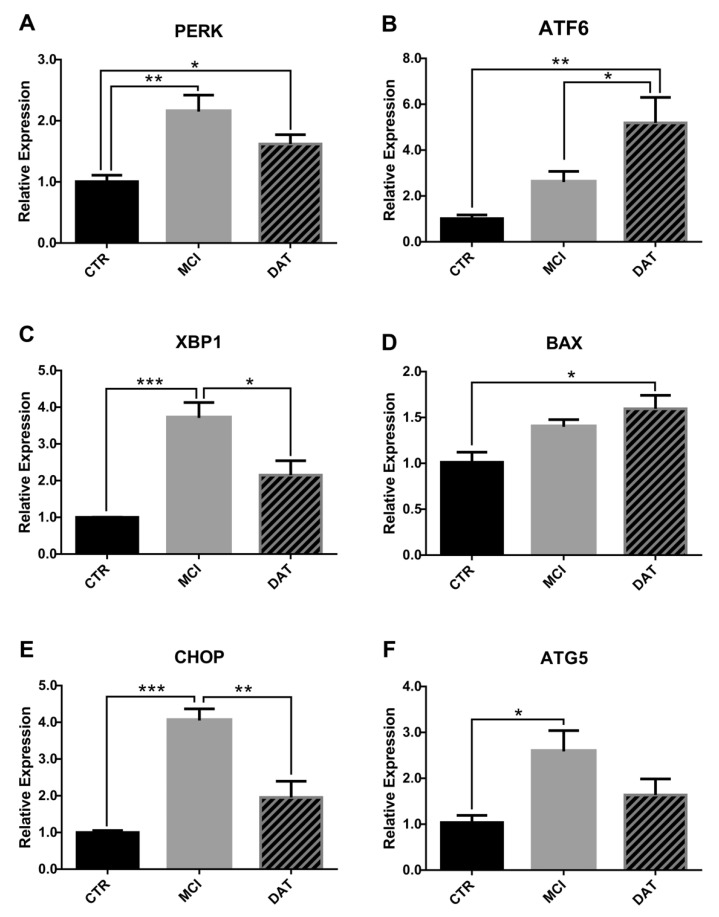
Relative expression analysis of molecular markers associated with morphological alterations. The status of RER and mitochondria was assayed in leukocytes by qRT-PCR analysis of specific molecular markers: (**A**) *PERK*, (**B**) *ATF6*, (**C**) *XBP1*, (**D**) *BAX*, (**E**) *CHOP*, and (**F**) *ATG5*. *PERK*, *XBP1*, *CHOP*, and *ATG5* expression was significantly higher in MCI patients. Conversely, the levels of ATF6 and BAX transcripts were higher in DAT patients. Results were obtained from five MCI and five DAT patients with respect to five healthy CTR. ANOVA parametric test followed by the Least Significant Difference (LSD) post hoc test for multiple comparisons. * *p* < 0.05, ** *p* < 0.01, *** *p* < 0.005.

**Figure 8 ijms-24-07909-f008:**
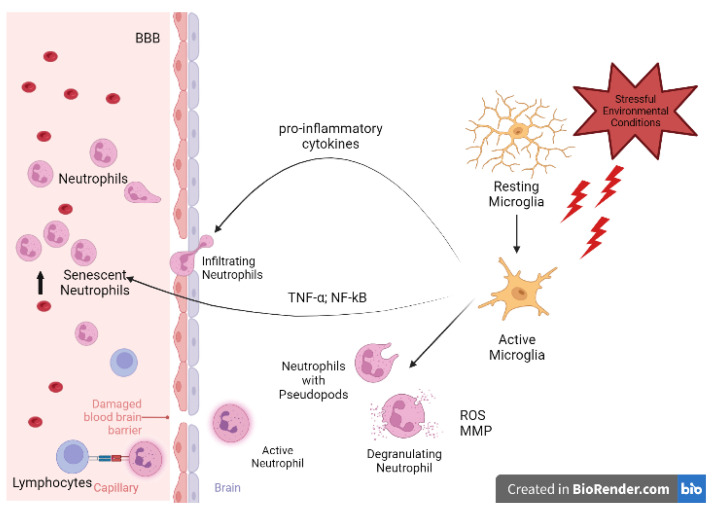
Neutrophil-to-microglia crosstalk. Stressful conditions promote microglia switching from a resting to a pro-inflammatory, cytokine-releasing active state aimed to enroll peripheral immune cells. Moreover, microglia produce TNF-α and NF-kB, which expands neutrophils’ lifespan, accounting for the NLR value increase. Meanwhile, neutrophils switch to their active state, exhibiting pseudopodia, degranulating, and secreting ROS and MMP, thus worsening BBB disruption. Active neutrophils operate reverse transmigration to act as antigen-presenting cells (APCs) to lymphocytes and link the innate and the adaptive immune responses. BBB, Brain–Blood Barrier; MMP, Matrix Metallo-Proteinases; NLR, Neutrophil-to-Lymphocyte Ratio; ROS, Reactive Oxygen Species.

**Figure 9 ijms-24-07909-f009:**
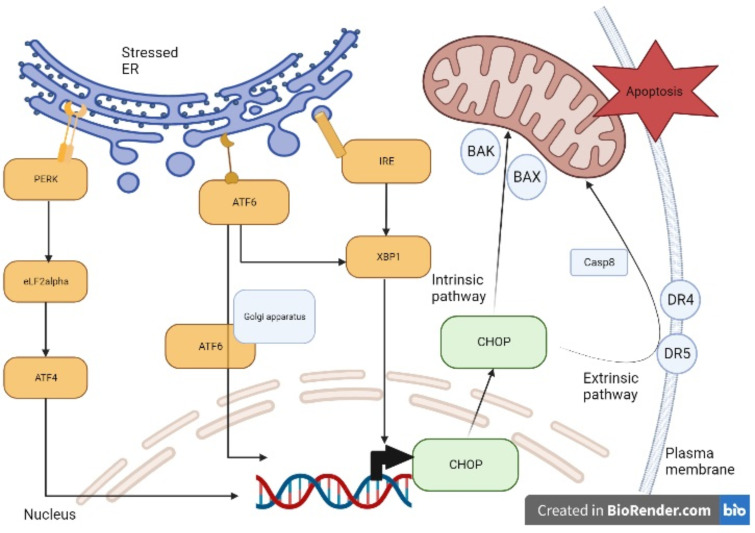
Stressed RER signaling and apoptosis. Under stressful conditions, the RER receptors *PERK*, *ATF6*, and *IRE1* are activated. *PERK* activates elf2alpha and, consequently, *ATF4* which translocates to the nucleus promoting *CHOP* expression. Once activated, *ATF6* translocates to the Golgi apparatus where it is cleaved and then it translocates to the nucleus where it contributes to *CHOP* expression as well. Similarly, *XBP1*, activated by *IRE1*, translocates to the nucleus promoting *CHO*P expression. *CHOP* induces the increased transcription of pro-apoptotic genes (*BAK* and *BAX*) involved in the intrinsic apoptotic pathway and of receptors (DR4 and DR5) involved in the extrinsic apoptotic pathway, which eventually activate caspase 8 (Casp8).

**Table 1 ijms-24-07909-t001:** Number of subjects, severity (CDR), and observations. We analyzed 51 MCI patients (CDR = 0.5–1), 84 DAT patients (CDR = 2–5), and 45 controls (CTR).

Subjects	CDR	Male	Observations	Females	Observations	Total Observations
CTR	0	16	32	29	58	90
MCI	0.5–1	21	40	30	63	103
DAT	2–5	30	61	54	134	195

**Table 2 ijms-24-07909-t002:** Comprehensive list of the human primers used for qRT-PCR analysis.

Gene	Sequence	Accession Number
*ATF6*	F 5′-AACCACTAGTAGTATCAGCAGGA-3′ R 5′-GGGGAGCCAAAGAAGGTGTT-3′	NM_007348.4
*ATG5*	F 5′-CAACTTGTTTCACGCTATATCAGG-3′ R 5′-CACTTTGTCAGTTACCAACGTCA-3′	NM_001286106.2
*BAX*	F 5′-TGACATGTTTTCTGACGGCA-3′ R 5′-CCA ATGTCCAGCCCATGATG-3′	NM_001291428.2
*CHOP*	F 5′-CATTGCCTTTCTCCTTCGGG-3′ R 5′-CCAGAGAAGCAGGGTCAAGA-3′	NM_001195053.1
*GAPDH*	F 5′-AGCCACATCGCTCAGACA-3′ R 5′-GCCCAATACGACCAAATCC-3′	NM_002046.7
*NF-kB*	F 5′-AATGGTGGAGTCTGGGAAGG-3′ R 5′-TCTGACGTTTCCTCTGCACT-3′	NM_003998.4
*NFKBIA*	F 5′-GTTGAAGTGTGGGGCTGATG-3′ R 5′-GTCCTCTGTGAACTCCGTGA-3′	NM_020529
*PERK*	F 5′-GGATCCGTCTCCCCAATAGG-3′ R 5′-GGCCAGTCTGTGCTTTCATC-3′	NM_004836.7
*TNF-α*	F 5′-AGGACCAGCTAAGAGGGAGA-3′ R 5′-CCCGGATCATGCTTTCAGTG-3′	NM_000594.4
*XBP1*	F 5′-CCGGAGCTGGGTATCTCAAA-3′ R 5′-GGCAAAAGTGTCCTCCCAAG-3′	NM_001079539.2

## Data Availability

The data that support the findings of this study are available in IFC-CNR and in Department of Experimental and Clinical Medicine, University of Pisa.

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
