# Peer review of "Ultrastructural and Molecular Investigation on Peripheral Leukocytes in Alzheimer’s Disease Patients"

_ijms, 2023, doi:10.3390/ijms24097909_

Round 1

Reviewer 1 Report

In this paper, the authors investigated the realtionship of alterations in peripheral leukocytes with Alzheimer's disease. They found that the number of neutrophils increased in both mild cognitive impairment (MCI) and demetia  (DAT) patients. Gene expression analyses revealed that neutrophil survival-related markers, such as TNF-alpha, NF-kB, and NFBIA, increased in affected people, especially, in MCI groups. TEM-analyses showed that rough endoplasmic reticulum (RER) altered in leukocytes derived from MCI and DAT patients.

The results of this study are intriguing and support current findings that  peripheral inflammatory responses are also involved in ALzheimer's disease pathogenesis.

In Figure 5, there were no differences in the number of autophagic vacuoles among three groups. However, in Figure 7, the expression level of ATG5, the master regulator of autophagy, was significantly upregulated in MCI patients. The authors should discuss more on this point. Sine recent findings suggest that autophagy can affect the secretion of extracellular vesicles, the upregulation of ATG5 may alter secretory pathway of leukocytes.

Author Response

Reviewer 1

Question 1. “In this paper, the authors investigated the realtionship of alterations in peripheral leukocytes with Alzheimer's disease. They found that the number of neutrophils increased in both mild cognitive impairment (MCI) and demetia  (DAT) patients. Gene expression analyses revealed that neutrophil survival-related markers, such as TNF-alpha, NF-kB, and NFBIA, increased in affected people, especially, in MCI groups. TEM-analyses showed that rough endoplasmic reticulum (RER) altered in leukocytes derived from MCI and DAT patients.

The results of this study are intriguing and support current findings that peripheral inflammatory responses are also involved in ALzheimer's disease pathogenesis.

In Figure 5, there were no differences in the number of autophagic vacuoles among three groups. However, in Figure 7, the expression level of ATG5, the master regulator of autophagy, was significantly upregulated in MCI patients. The authors should discuss more on this point. Sine recent findings suggest that autophagy can affect the secretion of extracellular vesicles, the upregulation of ATG5 may alter secretory pathway of leukocytes.”

Answer 1.

In agreement with Reviewer 1, we further discuss this point in the Discussion [lines 382-390]:

“Furthermore, the fact that there are no significant differences in the number of autophagic vacuoles among the three groups and that the expression level of ATG5, the master regulator of autophagy, is significantly upregulated in MCI patients, could induce to speculate the existence, also in leukocytes, of a secretory autophagy pathway (unconventional non-degradative process) involved in the recruitment of the cargo and the secretion of extracellular vesicles (EVs) as it has recently been reported in other mammalian cells (Eleni-Myrto Trifylli et al. 2022; Leidal and Debnath 2021). Moreover, at the disease onset in MCI, the EVs produced by leukocyte secretory autophagy could contribute to the activation of further pathogenetic mechanisms and therefore lead to AD progression.”

Reviewer 2 Report

The paper from Giannelli et al. investigates the role of peripheral leukocytes in neuroinflammation and neurodegeneration, as key actors of the detrimental immune response in AD patients. The work is pretty interesting and well written but i do have some major concerns. 

1) It is not clear to me why author choose to use two different graphical approaches to present their data. For example, we find boxplots in figure 1 and 2, but they use histograms in all other figures. Can authors please explain this choice in the "statistical analysis" chapter?

2) Since authors, in the last chapter of materials and methods, explain that they consider different p values, different parametric/non-parametric tests etc., it would be easier for the reader to understand presented data if the statistical tests used for each figure was clearly reported in the figure legend. Please add these details and every more detail that would help the understanding.

3) In some data sets, SEM is quite high (fig. 4E, 5D). Would it be possible to provide dispersion of data? Both adding a supplementary file with raw data or adding dots to represent every single measurement to the figures would help.

4) Authors choose to evaluate expression levels of different proteins measuring their transcripts. What about actual protein level? It is not always true that an increase in protein's transcript results in a consequential increase in protein itself. Can author please add some more molecular analysis in this regard? Western blot or ELISA experiments would be very good to confirm an actual increase/decrease of a target protein.

5) It is not clear to me how Alzheimer's disease fits in this story. Authors consider male and female patients evaluated by their CDR, divided in MCI, DAT and CTR subjects. A little paragraph in the discussion chapter, which is: "In our research, we collected blood samples from patients with slight cognitive impairment functionally indistinguishable from those of age-matched control subjects and stocked them in a biobank. This allows us to analyze the blood samples stocked several years prior to AD diagnosis retrospectively and, thus, focus on the occurrence of the illness." is, in my opinion, very short and do not help the reader to understand author's vision of the story. Furthermore, how do authors exclude that the same changes in neuroinflammatory pathways can occur in the early phases of other neurodegenerative diseases (i.e. Parkinson's disease). Please add a substantial chapter in the discussion/conclusions chapter to better explain.

Minor concerns:

1) Can authors add a little more insights about the "Leukocyte preparation" chapter? Any more information on the Lympholite solution?

2) Figure 4E. Please adjust the y axis.

Author Response

Reviewer 2

Question 1. “The paper from Giannelli et al. investigates the role of peripheral leukocytes in neuroinflammation and neurodegeneration, as key actors of the detrimental immune response in AD patients. The work is pretty interesting and well written but i do have some major concerns.

It is not clear to me why author choose to use two different graphical approaches to present their data. For example, we find boxplots in figure 1 and 2, but they use histograms in all other figures. Can authors please explain this choice in the "statistical analysis" chapter?”

Answer 1.

The choice of boxplots as a graphical approach rather than histograms relies on the data distribution. For not normally distributed data we decided on boxplots since the non-parametric tests adopted for them are determined by the median represented in the box, whereas the parametric tests exploited for normally distributed data are generated starting from the mean value.

Question 2. “Since authors, in the last chapter of materials and methods, explain that they consider different p values, different parametric/non-parametric tests etc., it would be easier for the reader to understand presented data if the statistical tests used for each figure was clearly reported in the figure legend. Please add these details and every more detail that would help the understanding.”

Answer 2.

Details about the statistical tests have been added in the legend of the figures to help the understanding:

- Figure 1 Kruskall-Wallis non-parametric test followed by Mann-Whitney U-test for group comparison            

- Figure 2,7  ANOVA parametric test followed by the Least Significant Difference (LSD) post hoc test for multiple comparisons                                                                                                                          

- Figure 3,4,5,6  MANOVA followed by multiple range test (MRT) has been used for statistical analysis

Question 3. In some data sets, SEM is quite high (fig. 4E, 5D). Would it be possible to provide dispersion of data? Both adding a supplementary file with raw data or adding dots to represent every single measurement to the figures would help”

Answer 3.

In order to answer the request of Reviewer 2, we deepened the “Statistical Analysis” paragraph. We reckon a dispersion graph would not be the proper choice for the aspect we want to stress on, which is the comparison among groups. Graphs 4E and 5D, as well as all the TEM-linked data, follow a binomial distribution. As reviewer 2 rightfully indicated, we should have specified that TEM data are expressed as mean ± DS.

The “Statistical analysis” paragraph has been modified by adding the following information:

“For morphometric analysis by TEM, parametric variables are presented as mean ± DS. Data follow a Bernoulli binomial distribution as explained in “Morphometric analysis: assessment of rough endoplasmic reticulum, mitochondria, autophagic vacuoles and pseudopods”.

Question 4. “Authors choose to evaluate expression levels of different proteins measuring their transcripts. What about actual protein level? It is not always true that an increase in protein's transcript results in a consequential increase in protein itself. Can author please add some more molecular analysis in this regard? Western blot or ELISA experiments would be very good to confirm an actual increase/decrease of a target protein.”

Answer 4.

In the case of a functional study of Alzheimer's disease, a protein dosage of the genes found altered by qRT-PCR is, certainly, essential. The aim of our investigation, on the other hand, is to evaluate whether and to what extent leukocytes can be carriers of information on the onset of AD. To do that, we first performed ultrastructural analyses on leukocytes of AD patients with different levels of severity compared with healthy controls. Consequently, we analysed molecular markers associated with the observed morphological alterations. Taken together, these data could lead us to identify possible peripheral early biomarkers for AD.

Question 5. “It is not clear to me how Alzheimer's disease fits in this story. Authors consider male and female patients evaluated by their CDR, divided in MCI, DAT and CTR subjects. A little paragraph in the discussion chapter, which is: "In our research, we collected blood samples from patients with slight cognitive impairment functionally indistinguishable from those of age-matched control subjects and stocked them in a biobank. This allows us to analyze the blood samples stocked several years prior to AD diagnosis retrospectively and, thus, focus on the occurrence of the illness." is, in my opinion, very short and do not help the reader to understand author's vision of the story. Furthermore, how do authors exclude that the same changes in neuroinflammatory pathways can occur in the early phases of other neurodegenerative diseases (i.e. Parkinson’s  disease). Please add a substantial chapter in the discussion/conclusions chapter to better explain.”

 Answer 5.

The paragraph has been modified according to the Reviewer suggestions [lines 272-279]:

“Another critical issue raised by various researchers is that NLR is generally calculated when the patients' CDR is at least 1, since before it is not possible to distinguish an AD onset from a physiological cognitive decline due to aging [14]. This study aims to observe how peripheral leukocyte features change mainly at the onset of AD to identify potential molecular markers when the first events of pathological neurodegeneration arise. We collected blood samples from patients with slight cognitive impairment functionally indistinguishable from those of age-matched control subjects and stocked them in a biobank even several years before the patient is diagnosed with a neurodegenerative disorder. This allows us to analyze retrospectively the samples when the disease is established and, thus, focus on the first alterations with the illness occurrence.”

About the final portion of question 5 (“Furthermore, how do authors exclude that the same changes in neuroinflammatory pathways can occur in the early phases of other neurodegenerative diseases (i.e. Parkinson’s  disease). Please add a substantial chapter in the discussion/conclusions chapter to better explain.”), we stated that, indeed, RER has been shown to be involved in several neurodegenerative diseases:

[lines 366-368]

“Based on our results and considering that the presumed contact of leukocytes with an inflamed brain environment may alter their morphology, the stressed RER may be a signal of neurodegeneration for AD, as also suggested by Merighi et al. [25].”

Nevertheless, a detailed clarification on our aim regarding RER investigation is missing.  We are interested in a diagnostic test discriminating between AD patients and healthy elderly subjects. A further explanation has been added in “Materials and Methods, Patients enrolled”.

To better convey this concept, an extensive paragraph have been added in the text:  

[lines 368-375]

“Indeed, RER stress has been demonstrated to be involved in different neurodegenerative dis-eases, such as Parkinson Disease and Amyotrophic Lateral Sclerosis [25]. Therefore, we propose to distinguish AD onset from other neurological disorders basing on an integration of RER stress with more specific clinical traits. In addition, in our work we propose to use RER stress biomarkers in peripheral leukocytes to point out AD onset as compared with age- and sex-matched healthy controls after having excluded other neurodegenerative conditions.”

Minor concerns:

Question 1. “Can authors add a little more insights about the "Leukocyte preparation" chapter? Any more information on the Lympholite solution?”

Answer 1.

The following sentence has been added in the text (“Materials and Methods” section, lines 427-429):

“Lympholyte solution (Cedarlane, Burlington, Canada) has been used as a density gradient separation medium in a conic 15 ml tube for the isolation of mononucleated cells from the other blood cells.”

Question 2. Figure 4E. Please adjust the y axis”

Figure 4 has been modified according to the Reviewer’s suggestion.

Round 2

Reviewer 2 Report

I thank the authors for their answers. In my opinion, manuscript has been improved.